# Combination of Solid State and Submerged Fermentation Strategies to Produce a New Jellyfish-Based Food

**DOI:** 10.3390/foods11243974

**Published:** 2022-12-08

**Authors:** Francesca Anna Ramires, Gianluca Bleve, Stefania De Domenico, Antonella Leone

**Affiliations:** 1Consiglio Nazionale delle Ricerche, Istituto di Scienze delle Produzioni Alimentari, Unità Operativa di Lecce, 73100 Lecce, Italy; 2Dipartimento di Biologia e Scienze Biologiche e Ambientali (DiSTeBA), Campus Ecotekne, Università del Salento, 73100 Lecce, Italy; 3Consorzio Nazionale Interuniversitario per le Scienze del Mare (CoNISMa), Local Unit of Lecce, 73100 Lecce, Italy; 4National Biodiversity Future Center (NBFC), 90133 Palermo, Italy

**Keywords:** fermentation, edible jellyfish, safety assessment, quality traits, nutritional traits, odor profile, novel foods

## Abstract

This study describes the set-up and optimization of a fermentation strategy applied to a composite raw material containing jellyfish biomass as the principal ingredient. New fermented food was developed by combining fresh jellyfish *Rhizostoma pulmo* and the sequential solid-state submerged liquid fermentation method used in Asian countries for processing a high-salt-containing raw material. *Aspergillus oryzae* was used to drive the first fermentation, conducted in solid-state conditions, of a jellyfish-based product, here named Jelly paste. The second fermentation was performed by inoculating the Jelly paste with different selected bacteria and yeasts, leading to a final product named fermented Jellyfish paste. For the first time, a set of safety parameters necessary for monitoring and describing a jellyfish-based fermented food was established. The new fermented products obtained by the use of *Debaryomyces hansenii* BC T3-23 yeast strain and the *Bacillus amyloliquefaciens* MS3 bacterial strain revealed desirable nutritional traits in terms of protein, lipids and total phenolic content, as well as valuable total antioxidant activity. The obtained final products also showed a complex enzyme profile rich in amylase, protease and lipase activities, thus making them characterized by unique composite sensory odor descriptors (umami, smoked, dried fruit, spices).

## 1. Introduction

Since ancient times, fermentation has been used as a process generally able to modify and produce foods. A number of raw meats, fish, milk, grains, fruits and vegetables can be fermented, producing a variety of preserved foods [1]. They are currently increasing in popularity since fermentation changes the nutrient content, digestibility and sensory characteristics of these products [2].

To our knowledge, the fermentation process has not yet been applied to jellyfish as raw material, with the exception of a Chinese patent related to the application of lactic acid bacteria to alum-treated jellyfish for the production of a fermented version of this product [3] and in the study of Wang et al. [4], where the effects of the inoculation of *Lactobacillus* strains into desalted alum-treated jellyfish were studied to enhance the flavor of this product and reduce the use of other preservatives.

Asian countries are big producers and consumers of jellyfish, which represents a favorite and popular seafood. Indeed, jellyfish is regarded as a high-quality diet marine product [5]. In recent years, much attention has been paid to jellyfish, with the exploitation of new and more sustainable marine resources as food. The main products commercially available are instant and salted jellyfish, produced and exported from Eastern countries. These Asian traditional products can be preserved conveniently and kept for a long time. However, desalination is necessary before eating them. Instant jellyfish are simpler to eat than salted ones, but they have a limited shelf-life and cannot be preserved for a long time period. In fact, the preservation conditions are harsh and energy-demanding. Many studies explore instant jellyfish in China [6,7,8].

Although jellyfish are not a traditional food in Western countries and in Europe, an increasing interest in their use as food was reported [9,10,11,12,13]. In addition, the presence of bioactive compounds and valuable antioxidant and antiproliferative activities in some Mediterranean jellyfish species [14,15,16] makes them increasingly attractive.

Due to the high water content (about 95%) and rapid perishability of the jellyfish after fishing, fast processing is required in order to stabilize the biomass. Fermentation, largely developed for the treatment of agri-cultural products [17], was here explored as a food processing technology suitable for the stabilization and transformation of the highly unpreserved jellyfish raw material, by using a combination of the growth and metabolic activity of appropriate microorganisms and proper raw material processing conditions.

The here-proposed fermentation approach could offer an alternative for the management and valorization of the increasing presence and even abnormal accumulations (“blooms”) of jellyfish in several seaside zones. Indeed, they can be problematic for fishing operations, aquaculture plants and other anthropic activities in coastal areas [5].

However, since jellyfish body tissues have a total protein, lipid and carbohydrate content of 7.3 ± 3.6% DW [18], they cannot be directly used as a substrate for fermentation by themselves, but they can be proposed as an ingredient for fermented food preparation.

The aim of this study was to develop a new method to ferment jellyfish mixed with adequate vegetable sources of fibers and carbohydrates, by gathering and adapting those traditionally used for the production of fermented soybean, rice or barley paste, named miso (Japan), jiang (China) and doenjang (Republic of Korea). These are the basis of various dishes such as soup, stew and salad dressing [19].

A Koji inoculum represented by the fungal starter (especially *Aspergillus*) produces and secretes several useful enzymes [20], suitable for the subsequent fermentation step. Solid-state fermentation is then carried out by the Koji inoculum, in absence or very low free-flowing water. This technology, which resembles natural growing environments, offers to microbial sources (i.e., filamentous fungi) the possibility to directly metabolize substrates with minimal pretreatment, also when nutrients are in low quantities [21,22]. This approach also takes advantage of the on-site production of microbial hydrolytic enzymes to obtain easily digestible sugars, nitrogen and fatty acids by the break-down of the complex molecules [23].

The obtained product can afterward undergo a subsequent submerged liquid fermentation, normally preferred for processes driven by yeast and bacterial inocula, where the ingredients are commonly administered in low and diluted concentrations.

The traditional process for miso and soy sauce production is poorly controlled; later, well-defined mixed starter formulations were proposed, by both co-inoculation or sequential inoculation of bacteria and yeasts [20,24]. This strategy helps in maintaining or improving the development of a unique flavor, texture and nutritional profile [25,26]. Indeed, the fermentation process contributes to the high level of bioactive compounds such as essential amino acids, minerals, vitamins, phenolic compounds, isoflavones and saponins [27,28].

In this paper, for the first time, a sequential solid and submerged liquid fermentation strategy was developed and applied for producing jellyfish-based fermented food products, as described by Cunha et al. [29]. Several examples of successful applications of this procedure have been reported for numerous substrates and products [30].

The main goal of this preliminary study was the set-up and optimization of a fermentation strategy applied to a composite raw material containing jellyfish biomass as the principal ingredient. The newly proposed process consisted of (i) the preparation of Koji, (ii) the solid-state fermentation phase driven by Koji to obtain the intermediate product named Jelly paste and (iii) the subsequent submerged liquid fermentation step to produce the Jelly fermented paste. In this last step, several selected bacterial and yeast strains coming from different origins, including newly isolated and selected ones from the jellyfish species *Rhizostoma pulmo*, were tested as starters for their fermentation performances. The whole process was monitored, and safety, sensory characterization and nutritional aspects of the final products were evaluated.

## 2. Materials and Methods

### 2.1. Sample Collection

*Rhizostoma pulmo* jellyfish specimens were collected from an open boat on the coast of the Ionian Sea at Marina di Ginosa (Taranto, Italy) (40°24’36.8” N 16°53’04.0” E) with a 3.5 cm mesh nylon fishing net, during the samplings in the summer periods of 2018 and 2020. Alive jellyfish were temporarily stored at 10–15 °C, immersed in seawater in food-grade containers, and transported to the laboratory within three hours after collection. Since specific slaughter guidelines for cnidarians are not available [31], the traditional method to kill the jellyfish used in Asian countries [5] was applied to *R. pulmo* specimens.

In the laboratory, jellyfish were immersed in sterile seawater (SW) and the umbrella and oral arms were separated with a plastic knife to remove the content of the digestive cavity. The umbrella and oral arms were washed three times in sterile seawater and then submitted to treatments for jellyfish paste preparation or frozen at −80 °C in sterile bags for later treatment.

### 2.2. Microbiological Analyses

Microbiological analyses of jellyfish tissue were performed following the procedure described by Bleve et al. [9,10] with some modifications. Samples diluted with 1 g/L (*w*/*v*) peptone water were analyzed on agar media plates: for total bacterial count, Plate Count Agar (PCA, Heywood, Lancashire, UK) with 0.05 g/L nystatin (Sigma-Aldrich, Darmstadt, Germany) and incubated for 48–72 h at 30 °C; for Enterobacteriaceae, Violet Red Bile Glucose Agar (VRBGA, LABM, Heywood, Lancashire, UK) incubated at 37 °C for 18–24 h; for coli–aerogenes bacteria, Violet Red Bile Agar (VRBA, LABM, Heywood, Lancashire, UK) incubated at 37 °C for 24–48 h; for coagulase-positive staphylococci, Baird Parker Agar Base (BP, LABM, Heywood, Lancashire, UK) incubated at 37 °C for 24–48 h; for pathogenic staphylococci, Mannitol Salt Agar (MSA, LABM, Heywood, Lancashire, UK) incubated at 37 °C for 18–72 h; for *Vibrio* spp., Thiosulfate Citrate Bile Sucrose Agar (TCBSA, Sigma-Aldrich, Darmstadt, Germany) incubated at 37 °C for 18–24 h; for *Bacillus* spp., Bacillus ChromoSelect Agar (BCSA, Sigma-Aldrich, Darmstadt, Germany) with Polymyxin B supplement incubated at 30 °C for 24–48 h; for H_2_S producing bacteria, Iron Agar (Lyngby) without cysteine (Sigma-Aldrich, Darmstadt, Germany) incubated at 25 °C for 48 h; for *Pseudomonas* spp., Pseudomonas agar (LABM, Heywood, Lancashire, UK) added with CFC supplement (LABM, Heywood, Lancashire, UK) incubated at 35 °C for 24–48 h. The presence of yeast and molds was tested by incubation at 25 °C for 5 days on Dichloran Rose-Bengal Chloramphenicol Agar (DRBC, Thermo Fisher Scientific, Monza, Italy). For the determination of halophilic microorganisms, the procedure described by Bleve et al. [10] was followed.

The isolation of yeast and bacteria from jellyfish tissue was performed following the procedure described by Bleve et al. [9,10]. Based on morphological traits, representative isolated bacteria and yeasts were selected. The isolated bacteria were maintained on saline MRS agar or saline PCA added with 0.05 g/L nystatin (Sigma-Aldrich, Darmstadt, Germany). The isolated yeasts were maintained on saline Sabouraud dextrose agar (Sigma-Aldrich, Darmstadt, Germany) supplemented with 0.05 g/L ampicillin (Sigma-Aldrich, Darmstadt, Germany) at 28 °C for 2–3 days. Bacteria maintained on saline MRS agar were grown anaerobically at 28 °C for 3–4 days. Bacteria maintained on saline PCA were grown aerobically at 28 °C for 2–3 days.

### 2.3. Molecular Identification of Bacterial and Yeast Isolates

The bacterial total genomic DNA was extracted by using the Power Soil DNA Isolation Kit (MO BIO, Carlsbad, CA USA), and the 16S rDNA region was amplified according to Bleve et al. [32].

The total genomic DNA from the yeast strains was prepared according to the method used by [33]. The genomic region ITS1-5,8 S-ITS2 was amplified as described by Bleve et al. [33] with the following modifications: initial denaturation at 95 °C for 5 min, followed by 35 cycles consisting of 30 s at 95 °C, 30 s at 52 °C and 1 min at 72 °C, followed by a final extension at 72 °C for 10 min. The amplicons were purified and sequenced as previously described by Ramires et al. [34]. Sequences were analyzed by the Chromas program, version 1.45 (www.technelysium.com.au, accessed on 4 March 2022), and by the BLAST program for sequence alignment and comparison.

### 2.4. Jellyfish Puree Preparation

Jellyfish as previously treated (Section 2.1) were washed with drinking water, cut into cubes of about 1.5 cm and then homogenized in a blender for 5–10 min at maximum speed. The homogenized jellyfish were transferred into a beaker containing a stirring magnet and added (slowly) with corn starch 12% (*w*/*w*) and dried wheat bran 12% (*w*/*w*), previously sterilized. The mixture was heated under stirring until a thick consistency was obtained and then kept at 45 °C. The obtained product was called Jellyfish puree.

### 2.5. Fermentation Procedure for Jelly Paste Preparation

The procedure for jellyfish fermentation and Jelly paste preparation consisted of three main steps, each one entailing specific experimental phases shown in Figure 1 and Appendix A.

#### 2.5.1. First Step: Koji Preparation

On day 0, commercial *Carnaroli* rice (1 kg) was washed to eliminate dust and bran with 1.5 L of water and then drained with a strainer. Then, the rice was soaked in 1.5 L of water for 45 min at room temperature and properly drained again. Then, it was divided into 0.5 Kg aliquots in flasks and sterilized for 20 min at 121 °C. The sterilized rice was stabilized at 45 °C and then inoculated with the lyophilized powder of *Aspergillus oryzae* CNR-ISPA-LE 112019 strain (about 200 mg for 1 kg of rice). The sample was kept at 32 °C for 18 h (day 1).

On day 2, the rice samples were stirred in order to untangle the clumps. The temperature was changed to 40 °C. After 4 h of incubation, the samples were stirred and incubated again at 40 °C.

On day 3, the rice samples were stirred in order to untangle the clumps and maintained at 39 °C for a further 48 h.

On day 4, when the hyphae of the mold were rooted in the rice and seemed to have attached themselves to the rice grains, Koji preparation was considered completed. It was used as the Koji starter for the subsequent fermentation step.

#### 2.5.2. Second Step: Solid-State Fermentation of Jellyfish Puree by Koji to Obtain the Jelly Paste

The Koji starter from the previous step was added to the Jellyfish puree, prepared as previously described (Section 2.4), at a percentage corresponding to 17.5% (*w*/*w*) of the final volume, accurately mixed and then spread in a large flat container, covered with a sterile gauze and then with a lid with holes to ensure the gas transfer. This preparation was incubated at 40 °C for 48 h to obtain the jellyfish puree matrix fermented by Koji, and it was called Jelly paste.

#### 2.5.3. Third Step: Jelly Paste Fermentation with Selected Yeasts and Bacteria Strains

Jelly paste obtained in step 2 (about 150 g) was collected from the tray and aliquoted in a glass jar, added in a 1:1 ratio with bi-distilled water (150 mL), and sterilized at 121 °C for 20 min.

Yeasts and bacteria strains were individually inoculated as starters at a final concentration of about 10^7^ CFU/g (as already used for starter formulations in fermentation of vegetables from terrestrial and marine origin by Tufariello et al. [35], Maiorano et al. [36]) into the sterilized Jelly paste. The bacterial and yeast strains individually used as candidate starters, all belonging to the CNR ISPA-LE microbial collection, were *Bacillus amyloliquefaciens* MS3, *Lactiplantibacillus plantarum* C 180-11, *Leuconostoc mesenteroides* KT 5-1, *Saccharomyces cerevisiae* LI 180-7 and *Debaryomyces hansenii* BC T3-23. Bacterial and yeast strains isolated and identified from *R. pulmo* were *Staphylococcus pasteuri* SB26 (Table 1), *Metschnikowia* sp. Y1D and *Candida parapsilosis* YB51 (Table 2). Commercial lyophilized microbial preparations, kindly provided by Sacco srl (Cadorago, Italy), were SBM-11 (*Lactobacillus sakei*, *Staphylococcus carnosus* and *Staphylococcus xylosus*) and PROMIX-1 (*Staphylococcus xylosus*).

The fermentation was carried out for 20 days at 30 °C for Jelly paste samples inoculated with yeasts and bacteria, with the exception of those inoculated with SBM-11 and PROMIX-1, which were incubated at 37 °C. Three uninoculated samples were also produced as control of spontaneous fermentation process.

Analyses of salinity and pH were performed every 5 days until day 20 (end of the experiment) on the obtained fermented Jellyfish pastes. Salinity was measured by using a salinity refractometer for seawater and a marine aquaria 0–10% hydrometer with automatic temperature RHS-MR110 ATC (Agritechstore, Mori, Trento, Italy). pH was measured using a pH-meter (Hanna Instruments Italia Srl—Ronchi di Villafranca Padovana, Padova, Italy).

### 2.6. Nutritional Analyses

#### 2.6.1. Protein Content

Fresh fermented jellyfish samples were homogenized in a Waring^®^ laboratory blender (three pulses for 15 s in refrigerated conditions). Approximately two grams of each homogenized sample were suspended in 2, 4 or 8 mL of MilliQ water in order to obtain a homogeneous suspension.

The Bradford assay [37] was used to evaluate the total protein content in each fermented jellyfish sample, and was modified and adapted to a round-bottom 96-well microplate for the Infinite 200 PRO microplate reader (TECAN, Männedorf, Switzerland), using bovine serum albumin (BSA, Sigma-Aldrich, Darmstadt, Germany) as standard [15]. Means of at least 3 measurements from two independent experiments were considered.

#### 2.6.2. Antioxidant Activity

The antioxidant activity was evaluated in each fermented jellyfish sample by the Trolox Equivalent Antioxidant Capacity (TEAC) method, adapted for the Infinite 200 PRO microplate reader (TECAN, Männedorf, Switzerland) using the radical cation ABTS•+ and Trolox (Sigma-Aldrich, Darmstadt, Germany) as standard [38,39]. The samples and the standard were assayed under the same conditions as already described by De Domenico et al. [15]. Results were expressed as nmol of Trolox equivalent per g of fresh weight (nmol TE/g FW), and the means of at least 3 measurements from two independent experiments were considered.

#### 2.6.3. Lipid Extraction and Determination

Total lipids were extracted using the modified method of Bligh and Dyer [40] with some modifications [16]. Dried samples (200 mg) were mixed with a total of 15 mL solvent added in this sequence: 6 mL of chloroform:methanol (2:1), 6 mL of chloroform:methanol (2:1) and 3 mL KCl (0.88%). Samples were shaken for 15 s after the addition of each solvent and centrifuged at 5140× *g* for 5 min. The lower phase was set aside, and the upper phase was subjected to further extraction with a solution of chloroform:methanol (2:1, 1 *V*). The lower phase was isolated and added to the first one and mixed with a solution of methanol:water (1:1, ¼ *V*). In this case, the lower phase was put aside, dried in the presence of nitrogen flux and analyzed for total lipid quantification.

#### 2.6.4. Total Phenolic Content

The assay described by Magalhães et al. [41] was used to measure the content of phenolic compounds in fermented Jellyfish pastes. Homogeneous suspensions of samples were prepared by proper diluting, and an aliquot of 50 μL of suspension from each sample was added to 50 µL of Folin–Ciocalteu reagent. After mixing, 100 µL of NaOH 0.35 M was added to neutralize the reaction, and after incubation in the dark at room temperature for 5 min, the spectrophotometric reading at λ = 720 nm was performed. The blank was 50 µL of water added with 50 µL of Folin–Ciocalteu and 100 µL NaOH 0.35 M. The obtained absorbance values were interpolated with the standard curve made using solutions with known concentrations of gallic acid in the range of 2.5–40 mg/mL. Results were expressed as gallic acid equivalent (GAE) per gram of fresh weight (FW).

#### 2.6.5. Glucose and Sucrose Content

For the quantification of glucose and sucrose, an aliquot of 1.5 g of fermented Jellyfish paste samples was suspended in 3 mL of distilled water and incubated with shaking at 100× g for 1 h at 30 °C. Then, this mixture was centrifuged at 20,000× *g* for 15 min; subsequently, the obtained supernatant was used for the Glucose and Sucrose Colorimetric Assay Kit (Sigma-Aldrich, Darmstadt, Germany) according to the manufacturer’s instructions.

### 2.7. Enzymatic Activity Assays of Jellyfish Paste during Fermentation

Enzyme activity assays for α-amylase, protease, esterase and lipase were determined to evaluate the production of these enzymes by the microbial starters at the end of fermentation. All experiments were conducted in triplicate.

#### 2.7.1. Preparation of Crude Enzyme Solution from Jellyfish Fermented Paste

Crude enzyme solutions from Jelly fermented pastes were prepared according to the method of Lee et al. [42] with slight modifications. Briefly, 2.5 g of fermented Jellyfish paste samples were suspended in 5 mL of distilled water and incubated with shaking at 100× g for 1 h at 30 °C. Then, these mixtures were centrifuged at 8000× *g* at 4 °C for 15 min. The obtained supernatants were recovered as crude enzyme solutions.

#### 2.7.2. α-Amylase Activity Assay

The α-amylase assay was performed using 72 µL of crude sample enzyme solution and a reaction mixture consisting of 50 µL substrate solution (1% potato starch in 1 M phosphate buffer pH 7) and 93 µL of 1M phosphate buffer at pH 7. The reaction was carried out at 40 °C for exactly 10 min and then it was stopped by adding 714 µL of 0.1 M HCl. Finally, 15 µL of blocked reaction solution was collected and diluted in 185 µL of sterile double distilled water plus 50 µL of 0.005% iodine solution, and the absorbance was measured at 660 nm using a nanodrop (Thermo Fisher Scientific, Rodano, Milano, Italy). A standard starch curve was prepared using a 5 mM stock solution. One unit (U) of activity of α-amylase is defined as the amount of enzyme required to release 1 μmol of glucose reducing sugar equivalent per minute.

#### 2.7.3. Protease Activity Assay

Protease activity was tested following the method of Walter [43] and Moyano et al. [44] with the use of casein as a substrate, as proposed by Sigma-Aldrich Company. The hydrolysis of 0.66% (*w*/*v*) of casein in 50 mM Tris-HCl buffer at pH 8 was used to measure the protease activity. Extracts (50 μL) were mixed with 150 μL of casein substrate and incubated at 37 °C for 30 min; then, the reaction was blocked by the addition of 200 μL of 10% (*w*/*v*) trichloroacetic acid (TCA) and the mixture was centrifuged at 8000× *g* for 5 min. An aliquot (10 μL) of the obtained supernatant was mixed with 50 μL of 2 M sodium carbonate, 20 μL of Folin–Ciocalteu phenol reagent and 180 μL of water and incubated at 37 °C for 3 min. Protease enzyme activity was measured as a change in absorbance at 765 nm using a microplate reader. One unit of enzyme activity was expressed as 1 μmol of tyrosine min^−1^ mg protein^−1^.

#### 2.7.4. Esterase Activity Assay

The carboxyl ester hydrolase (esterase) activity was determined using a modified spectrometric method [45] by checking the hydrolysis of p-nitrophenylbutyrate (p-NPB) to p-nitrophenol at 37 °C for 5 min. The reaction mixture consisted of 10 mM p-NPB dissolved in acetonitrile and appropriately diluted crude enzyme solution. The release of p-nitrophenol was measured at 410 nm. One unit (U) of esterase activity was defined as the amount of esterase needed to release 1 μmol of p-nitrophenol per minute from p-NPB.

#### 2.7.5. Lipase Activity Assay

Lipase activity was detected by a spectrometric method using p-nitrophenyl palmitate (p-NPP) as substrate [46]. The substrate solution was prepared by suspending 0.0159 g of p-NPP, 0.017 g of sodium dodecyl sulfate and 1.00 g of Triton X-100 in a final volume of 100 mL distilled water. The assay mixture consisted of 2.5 mL of substrate solution, 2.5 mL of 0.1 M Tris-HCl buffer (pH 7.5) and 1 mL of crude enzyme solution properly diluted. The release of p-nitrophenol from p-NPP at 37 °C was monitored at 400 nm every 5 min. One unit (U) of lipase activity was defined as the amount of lipase required to release 1 μmol of p-nitrophenol from p-NPP in 1 min under the corresponding conditions.

### 2.8. Descriptive Sensory Analysis of the Fermented Products

For the preliminary characterization of the sensory properties of the jellyfish-based fermented products, a sensory panel was made up of seven women and seven men (ranging from 33 to 70 years old). The fermented product samples were proposed to the panelists in transparent plastic containers for screening under normal white light at 24 °C. Two preliminary sessions were run to select the best descriptors for several odor attributes, and a third session was conducted to identify the intensity of the selected attributes/descriptors on a seven-point intensity scale (0—none; 1—very delicate; 2—delicate; 3—delicate to moderate; 4—moderate; 5—moderate to intense; 6—intense; 7—very intense). The results were the mean values of the two sensory sessions.

Due to the preliminary nature of this study, and since the panelists were involved as volunteers in safe sensory tests limited to the perceptions of the only odor components of the samples by avoiding any possible physical contact with the products, ethical review and approval were waived for this study.

### 2.9. Statistical Analysis

All data represent the mean of three independent replicates (*n* = 3). Statistical analysis was based on one-way analysis of variance. Tukey’s post-hoc method was applied to establish significant differences among means (*p* < 0.05, *p* < 0.01, *p* < 0.001). All statistical comparisons were performed using Sigma-Stat, version 3.11 (Systat Software Inc., Chicago, IL, USA).

Principal component analysis (PCA) was used to compare important microbiological, chemical and biochemical parameters and sensorial traits associated with the fermented samples. All statistical analyses were carried out using XLSTAT software (Addinsoft Inc., Long Island City, NY, USA).

## 3. Results

In this preliminary study, the main goal was the set-up and optimization of a fermentation strategy applied to a composite raw material containing jellyfish biomass as the principal ingredient, as described in detail in Figure 1 and Appendix A. Fermentations were driven by selected microbial strains including, for the first time, some isolates from *Rhizostoma pulmo* jellyfish. The obtained products were characterized for some safety, nutritional and sensory aspects for future food applications.

### 3.1. Selection of Lactic Acid Bacteria and Yeast Strains as Candidates for Jellyfish Fermentation

In order to find possible microbial starters able to ferment jellyfish-based products, in this study, some endogenous bacteria and yeast associated with the jellyfish species *Rhizostoma pulmo* were isolated on saline media.

A total of 180 bacterial colonies were randomly selected, according to their different morphological aspects, from saline PCA, and 52 colonies were selected from saline MRS. Concerning yeasts, a total of 24 isolates were selected onto saline Sabouraud Dextrose Agar medium. According to their different morphological aspects, 29 bacterial colonies from saline PCA and saline MRS agar medium and six yeast colonies were randomly selected.

They were identified at the species level by molecular methods using genomic 16 S rDNA region for bacteria and ITS1-5,8 S-ITS2 for yeasts (Table 1 and Table 2). For this purpose, new protocols of colony PCR genomic DNA amplification and sequencing for bacteria, yeasts and molds were optimized as described in Section 2.

In addition to microbial isolates obtained from *R. pulmo* jellyfish, bacteria and yeast strains derived from other food matrices, available in CNR-ISPA Lecce microbial collections, as well as commercial strains, were selected based on their technological and food safety traits.

The bacterial starters used in this study were one bacterial strain belonging to *Staphylococcus pasteuri* SB26, selected among the isolates obtained from *R. pulmo* (Table 1); *Bacillus amyloliquefaciens* MS3, *Lactiplantibacillus plantarum* C 180-11 and *Leuconostoc mesenteroides* KT 5-1 strains, selected among isolates available in the CNR-ISPA microbial collection; and commercial preparations SBM-11 (*Lactobacillus sakei, Staphylococcus carnosus and Staphylococcus xylosus*) and PROMIX-1 (*Staphylococcus xylosus*).

Yeast isolates belonging to *Metschnikowia* sp. Y1D and *Candida parapsilosis* YB51 were selected among isolates obtained from *R. pulmo* (Table 2). *Debaryomyces hansenii* BC T3-23 and *Saccharomyces cerevisiae* LI 180-7 strains were selected among isolates available in the CNR-ISPA microbial collection.

### 3.2. Set Up of Fermentation Conditions

A procedure combining both solid-state and submerged fermentation of jellyfish-based food was elaborated to set up the possible treatment of jellyfish as a raw material for the production of new fermented products. In the first step (Koji preparation, Figure 1 and Appendix A), the activity of the *Aspergillus oryzae* (CNR-ISPA 112019) strain as a starter was verified by the production of a sweet aroma (data not shown). On day 4, Koji preparation was considered completed when the hyphae were evidently rooted and attached to the rice grains and the Koji starter produced a chestnut-like aroma. In this step, the Jellyfish puree, consisting of homogenized jellyfish added with corn starch (12% *w*/*w*) and sterilized wheat bran (12% *w*/*w*), was also formulated.

As described in Section 2.4, in the second step of the procedure, the Koji starter was used to carry out the fermentation of the Jellyfish puree to obtain the Jelly paste product.

In the third step, Jelly paste derived from the second step was used as the substrate for the fermentation with selected yeasts and bacterial strains to obtain a product named fermented Jellyfish paste. Before the inoculation with the different microbial strains, Jelly paste obtained in step 2 was sterilized in order to strongly reduce or eliminate the bacteria and fungi counts (Figure 1).

The most important safety parameters followed in this study, selected in compliance with most important safety standards in Europe, Australia and the USA for similar foods, were applied to the newly proposed fermented Jellyfish paste and are reported in Table 3.

The uninoculated Jelly paste was used for the control spontaneous fermentation (SF). Indeed, it underwent a spontaneous fermentation driven by the residual microbial species persisting after the heat sterilization treatment (Figure 1).

After 20 days of incubation at 28–30 °C, among tested yeast strains, *C. parapsilosis* YB51 and S. cerevisiae LI 180-7 strongly reduced or eliminated the total bacterial (TBC) and *Bacillus* spp. counts (BAC) during fermentation, inhibited the development of coliforms, *Enterobacteriaceae*, *E. coli* and staphylococci (PST) (Figure 2). The ability of these yeast strains to effectively drive the fermentation was demonstrated by the increasing load of the yeast total count (YsSDA, Figure 2) during the process and by the high count obtained at the end of it.

The use of the *Metschnikowia* sp. Y1D strain produced a favorable environment for the development of a potentially undesired bacterial community (*Bacillus* spp. and presumptive coagulase-positive staphylococci, BAC and CPS), whereas a decrease in potential lactic acid bacteria was reported.

In the sample inoculated with *Debaryomyces hansenii* BC T3-23 strain, yeasts (YsSDA) showed a suitable survival level until the 10th day of fermentation and seemed to strongly favor the growth of lactic acid bacteria and staphylococci (MGS and PTS).

Among bacterial strains, the use of *Lactiplantibacillus plantarum* C180-11 strain induced a high survival level of lactic acid bacteria (MGS) after ten days of fermentation (about 10^7^ CFU/g), after which the count declined. The treatment with the C180-11 strain revealed also satisfactory levels of total bacterial count (TBC) (<10^4^ CFU/g), the absence of *Escherichia coli*, *Enterobacteriaceae* (VRBGA) and coliforms (VRBA), a level of presumptive *Bacillus cereus* <10^2^ CFU/g (BAC), and the absence of staphylococci (CPS) and molds/yeasts (YDRBC and MDRBC) (Figure 3).

The same behavior was reported by the use of the *L. mesenteroides* KT-5-1 strain. Here, lactic acid bacteria (MGS) survived at a high level after ten days of fermentation (about 10^7^–10^8^ CFU/g). The use of this strain produced satisfactory levels of total bacterial count (TBC), whereas a complete absence of E. coli (VRBA), *Enterobacteriaceae* (VRBGA), staphylococci (CPS) and molds/yeasts (YDRBC and MDRBC) was revealed (Figure 3). Only the count of *Bacillus* spp. (BAC) was higher than the limit of 10^4^ CFU/g, even though none of the observed colonies showed a phenotype ascribable to *B. cereus*.

The application of two commercial starter inocula (SBM-11 and PROMIX-1) was able to maintain the total bacterial count (TBC) and *Bacillus* spp. (BAC) level under the established limits; no molds/yeasts (YDRBC and MDRBC), *Enterobacteriaceae* (VRBGA), E. coli or coliforms (VRBA) were detected. Staphylococci (CPS) were revealed only in the SBM-11-treated sample, even though the related counts were always within the expected acceptable limit (Table 3), and coagulase-positive colonies were completely absent (Figure 3).

### 3.3. Chemical–Physical, Nutritional and Sensorial Characterization of the Fermented Products

#### 3.3.1. Compositional Analyses

The characterization of fresh jellyfish as a raw material was performed and the results were in the range of values already found in previous works [9,16,19]. In particular, the total proteins were in the range of 2.4 to 4 mg/g of fresh weight and lipids were in the range of 0.2–0.3 mg/g of fresh weight, while carbohydrates represented minor and often negligible components of the jellyfish tissues.

The characterization of the Jelly paste preparation, obtained by the Koji-driven fermentation of the Jellyfish puree and then sterilized, revealed the following composition: initial values in moisture of 713.74 ± 31.08 mg/g FW, total lipid content of 128.53 ± 10.33 mg/g FW, 10.58 ± 0.33 mg/g FW of proteins, sugars value of 3.42 ± 0.02 mg/g FW (as the sum of glucose and sucrose) (Table 4), salinity of 13% (*w*/*w*) and pH of 5.23.

The sterilized Jelly paste was used as a starting material for the subsequent fermentation by yeast and bacteria strains. Analogously to the microbial analyses, the evolution of the main chemical–physical parameters was followed during the whole fermentation process (20 days) (Table 5).

The use of four bacterial strains (C 180-11, KT 5-1, SBM-11 and PROMIX-1) produced an adequate decrease in the pH value (pH < 4.3) just after 5 days of fermentation, which can be a satisfactory condition for the safety of fermented products (Table 5). The final pH values for these fermented samples ranged between 3.23 and 4.15 on the 20th day of fermentation. However, the pH levels obtained by the two samples inoculated with the strains MS3 and SB26 were considered unsatisfactory, since they were higher than the value 4.3.

In the samples inoculated with the yeast strains, the pH values were all more than 4.3 throughout the entire fermentation period (Table 5). However, the LI 180-7 and YB51 starters showed the ability to influence the process by increasing the yeast load during the fermentation and at the end of it (Figure 2). Temperature and salinity (%) did not show substantial changes during the process time and among the all the jellyfish-based products (Table 5).

#### 3.3.2. Nutritional Analyses

A preliminary nutritional characterization of the Jelly fermented paste products was carried out by analyzing reducing sugars, total lipids, proteins and total phenol content and antioxidant activity in all the samples (Table 4). The results indicated that Koji (containing *A. oryzae*) treatment to produce Jelly paste liberated reducing sugars corresponding to 3.42 ± 0.02 mg/g FW substrate, thus producing a saccharification percentage, calculated by applying the equation of Spano et al. [52] and Mahamud and Gomes [53] corresponding to an initial value of 52%.

Since the best microbial performances were obtained during the first 10 days of the third step of fermentation (Jelly fermented pastes inoculated with selected yeasts and bacteria strains), both the nutritional and sensorial analyses of the fermented products were carried out at this time point. Then, as resumed in Figure 1 and Appendix A, the complete process to produce Jelly fermented pastes lasted four days (day 0 to day 3) for the first step to obtain the Koji starter, other three days (from day 4 to day 7) to perform the first fermentation driven by Koji starter (solid-state fermentation) to obtain the Jellyfish paste sample, and finally ten days (from day 7 to day 17) to carry out the second fermentation driven by bacterial and yeast starters to produce the Jellyfish fermentation pastes.

The amount of reducing sugar (glucose) content decreased in all the samples, as expected by the growth of microorganisms, with the exception of those inoculated with the bacterial strain MS3 and the yeast strain BC T3-23 (Table 4).

The protein content decreased in all analyzed samples due to the decomposition of initial value over the fermentation period (Table 4), whereas the moisture of all the samples tended to significantly increase during the fermentation period (*p* < 0.05) (Table 4).

Moreover, the preliminary results reported in this study revealed that the fermented Jellyfish pastes can be a suitable resource of lipids, even though a decrease in the initial total lipid content was observed. However, the fermented products showed a final content with a range of 19.89 ± 2.10 mg/g FW to 169.48 ± 9.88 mg/g FW of total lipids, indicating wide variability in the metabolic activities of the different strains on this class of macromolecules.

The content of total phenols in fresh *R. pulmo* jellyfish was in the range of 45–60 μg GAE/g of fresh weight, in agreement with previous results [11]. A decrease in the phenol content in Jelly paste and in all the bacteria- and yeast-inoculated Jelly fermented paste samples was observed (Figure 4a) in comparison with the Koji (*A. oryzae*)-treated Jelly paste (Figure 4a). The samples inoculated with the bacterial strain MS3 and the yeast strain BC T3-23 maintained the highest levels of these compounds, followed by that treated with the bacterial strain KT 5-1. In this last case, the phenol content was comparable with the SF sample.

A similar trend to the phenol content was observed in the antioxidant activity (AA), expressed as nmol of Trolox equivalent (TE) per gram of FW. Considering that the antioxidant activity in raw untreated material of *R. pulmo* was about 800 ± 164 nmol TE/g FW, in agreement with Leone et al. [11], an increase in AA was detected in Jelly fermented paste, probably due to the fermentation metabolism with the formation of antioxidant metabolites. This is also in agreement with De Domenico et al. [15], where the digestion of proteins produced small peptides with higher antioxidant activity than undigested proteins. The antioxidant activity can be related to various chemical species that are formed during the fermentation processes. A similar behavior was observed in AA in all Jelly fermented paste samples, with the exception of the one inoculated with the yeast strain BC T3-23 (10226 ± 164 nmol TE/g FW), which, after 10 days of fermentation, maintained the starting AA value (10703 ± 263 nmol TE/g FW) of the Jelly paste (only treated with *A. oryzae*) (Figure 4b). Moreover, also in this case, the samples treated with the bacterial strain MS3 and the yeast strain KT 5-1, as well as the SF sample, conserved a suitable level of AA at the end of the process, corresponding to 6330 ± 347 nmol TE/g FW, 5279 ± 328 nmol TE/g FW and 6357 ± 221 nmol TE/g FW, respectively (Figure 4b).

#### 3.3.3. Enzymatic Activity Assays

Enzyme activity assays for α-amylase, protease, esterase and lipase were determined in the fermented Jellyfish paste products (Table 6).

Among all inoculated samples, the one treated with the yeast *D. hansenii* BC T3-23 strain revealed the highest lipase, α-amylase and protease activities.

In comparison with the spontaneous fermentation (SF), the other yeast strains treated samples revealed individual enzyme profiles: the use of *S. cerevisiae* LI 180-7 produced higher lipase and esterase and lower α-amylase and protease activity; the use of *C. parapsilosis* YB51 produced higher lipase and protease activity; the inoculation of *Metschnikowia* sp. Y1D yielded higher lipase, esterase and protease activity (Table 6).

All bacteria-inoculated samples revealed higher lipase and esterase activities in comparison with the SF sample (Table 6). The samples treated with commercial bacterial preparations containing staphylococci (SBM-11 and PROMIX-1) also disclosed significant α-amylase activity, whereas SBM-11 and *Leuconostoc mesenteroides* KT 5-1 showed high protease activity.

#### 3.3.4. Descriptive Analysis of the Odor Sensory Profile

The present study cannot be compared with previous ones carried out on jellyfish, because our objective was to use jellyfish as a possible new ingredient for the production of fermented products. Since no previous tradition exists for these products, there was no standard to follow.

In the absence of established descriptors for the sensory attributes of the newly obtained fermented Jellyfish pastes, the preliminary odor sensory evaluation of these samples was performed as a preliminary test to identify the most promising ones characterized by acceptable flavor traits.

Odorous descriptors detected in all analyzed samples with similar sensory descriptors were grouped into twelve sensory descriptors during the preliminary session by the panelists.

The terms selected to describe the perceivable odors are reported in Table 7.

The sensory evaluation of the fermented samples revealed the different flavor profiles of the obtained products (Figure 5a,b). In all tested samples, the undesired moldy trait was detected only at very low levels. The undesirable aspects of baked, smoked and fermented were mainly associated with the Jelly paste and SF (spontaneous fermentation) samples and with those treated with the yeast strains Y1D, LI 180-7 and YB51. On the other hand, all the desirable odors (umami, dried and red fruits, spices) and the value of the overall odor intensity were detected with higher scores in samples treated with the bacterial *B. amyloliquefaciens* MS3 strain and the yeast strain *D. hansenii* BC T3-23 (Figure 5a,b).

#### 3.3.5. Principal Component Analysis

Principal component analysis was then applied to the main important microbiological, chemical and biochemical parameters and sensorial traits associated with the fermented Jellyfish samples.

In the bi-plot concerning jellyfish-based products fermented with different microbial strains, the total variance of the two main components was 60.34% (Figure 6). PC1 clustered samples treated with the bacterial strain MS3, the yeast strain BC T3-23 and spontaneous fermentation (SF), showing the Jelly paste (treated with the Koji containing the fungus *A. oryzae*) on the negative semi-axis of the first component, discriminating them from all the other treatments. The clustered group of MS3, the yeast strain BC T3-23 and SF samples were clearly located in the portion of the plane characterized by the pH, antioxidant activity (AA), total phenolic content, LAB count, amylase and lipase activities, and several of the desired odor traits. The second (C 180-11, SBM-11, PROMIX-1, LI 180-7, YB51, Y1D) and the third (KT 5-1 and SB26) groups treated with all the other selected bacteria and yeast starters were located in the opposite portion of the plane, mainly associated with a limited number of variables (moisture, yeast total counts, the fermented undesired sensory trait, esterase activity and salinity).

## 4. Discussion

Rhizostomeae jellyfish are attracting interest as a food resource also in Western countries due to both its biochemical composition and abundance. Edible jellyfish, a novel food containing protein (mainly collagen) and minerals, very low in fat and calories, have been explored as a new source to produce ingredients for fermented functional foods [16,54]. The feasibility of the use of dry salted jellyfish as an ingredient for the production of nutritionally enriched highly demanded products, such as crackers, snacks and edible gelatin, was recently demonstrated, especially in terms of antioxidant properties [55,56].

In this paper, untreated fresh jellyfish (*Rhizostoma pulmo*) was used as a raw material for formulating a new fermented product named fermented Jellyfish paste obtained by the application of a sequential solid-state and submerged liquid fermentation process. Similar processes were reported for bioconversion approaches to prepare proteins, organic acids, ethanol and microbial enzymes from different feedstocks, such as wheat by-products, vegetable meal and food wastes [30]. The procedure here developed is also an adaptation of the already successfully applied scheme for the production of modern high-salt, liquid-state fermentation based on the Japanese-style soy sauce brewing process [57].

*Aspergillus oryzae* demonstrated in this study its ability to synthesize amylases and proteases, but also lipases in the solid-state fermentation of a complex matrix containing jellyfish (Table 4, Jelly paste). Indeed, Koji (including *A. oryzae*) activity on Jelly puree to produce Jelly paste liberated reducing sugars and provided an increase in the saccharification percentage.

The fungal-treated material, a hydrolysate highly enriched in enzymes and simple compounds, was further used as a substrate for the subsequent application of selected bacteria and yeasts starters able to perform a submerged liquid fermentation. This last step has been widely used in soy sauce and miso production for flavor enhancement and stabilization of the final products [20,24,30]. In order to find possible microbial starters to ferment jellyfish-based products, in this study, some bacteria and yeast associated with jellyfish species *Rhizostoma pulmo* were isolated and identified for the first time. At present, the isolation and characterization of microorganisms associated with jellyfish, with the scope to be used as a source for further application, have been poorly explored [58,59]. The bacterial strain *Staphylococcus pasteuri* SB26 and the yeasts *Metschnikowia* sp. Y1D and *Candida parapsilosis* YB51 were chosen among those isolated from *R. pulmo.*

*Staphylococcus* has been detected to be one of the most abundant genera in the production of soy sauce fermentations [24]. However, to the best of our knowledge, no data concerning the use of coagulase-negative staphylococci as non-conventional starter cultures for fermenting Koji-based products have been reported. This last microbial group has recently been included in commercial starter preparations to produce animal-derived fermented food preparations based on fish, cheese and meat [60]. Additionally, the use of *Staphylococcus* starter strain for fermented Jellyfish paste production can be promising since a high bacterial abundance of this genus was also reported among the bacterial communities associated with doenjang fermentation. Moreover, *Staphylococcus* species are already considered starters for the fermentation of foods, including doenjang, sausage and fish sauce [61].

Among the other bacterial strains chosen as candidate starters was *Leuconostoc mesenteroides*. Although isolates belonging to this species were identified as a minor class in Koji samples and Chinese soy sauce brine [25,62], they were used in several food fermentation applications [35] and obtained the qualified presumption of safety (QPS) by the European Food Safety Authority [63]. Concerning *Lactiplantibacillus plantarum* strains, they are largely employed in several fermented foods, especially vegetables in the presence of high salt concentrations [64,65]. They are used for the fermentation of different vegetable matrices, and here-investigated lactic acid bacteria strains (*L. plantarum* C 180-11 and *Lc. mesenteroides* K T5-1) already showed their ability to adapt and survive under difficult environmental constraints [64,65].

The bacterial *B. amyloliquefaciens* MS3 strain was tested here for Jelly paste fermentation due to both the high frequent detection of this species during soy sauce fermentation and the successful application of strains belonging to this species as a highly salt-tolerant and beneficial starter microorganism [66].

Among yeasts as starters for the step of submerged liquid fermentation, isolates from the species *Candida parapsilosis*, *Debaryomyces hansenii*, *Metschnikowia* sp. and *Saccharomyces cerevisiae* were tested. Although at low frequency, the presence of *C. parapsilosis* and the genus *Debaryomyces* spp. were found in the microbiota composition of Japanese Koji samples [25].

*C. parapsilosis* YB51 strain in fermented Jellyfish paste seemed able to reduce the development of potentially undesired bacteria, contrarily to what was observed for coffee fermentation [67]. Additionally, the *C. parapsilosis*-inoculated sample showed high lipase and protease activities, thus confirming that isolates belonging to this species can have potential to be used as starter cultures, as already tested in high-salt, liquid-state moromi to promote soy sauce fermentation [68]. However, the use of *C. parapsilosis* strains for food fermentations needs to be evaluated case by case.

The non-conventional yeast species *D. hansenii* was chosen since it was already isolated from saline water, foods, fruits and the human gut, and it is halotolerant and carries the Qualified Presumption of Safety status by the European Food Safety Authority, considering its long history as food yeast [69]. The *D. hansenii* BC T3-23 strain was able to control the process, also strongly supporting the growth of lactic acid bacteria and staphylococci. This last evidence is in accordance with the reported probiotic properties of *D. hansenii*: viability in the presence of bile salts and at low pH, immunomodulatory activity, increase/regulation of intestinal microflora, promotion of lactase-producing bacteria and avoidance of opportunistic pathogens in mouse model [70,71]. The indication that *D. hansenii* BC T3-23 can be a promising starter candidate was also confirmed by the peculiar enzyme profile associated with the fermented Jellyfish paste treated by this strain. The high α-amylase activity registered in this sample is a further verification of the possible use of this species as a new probiotic [72]. Moreover, the highest protease activity could also explain the recorded high antioxidant activity value, possibly linked to the production of *R. pulmo* small antioxidant peptides [15]. *D. hansenii* species were isolated as the most abundant yeast species during fermentation of Korean fermented soy sauce [73], with biotechnological importance in producing various volatile flavor compounds and enzyme patterns [74].

Concerning the other yeast species, *Metschnikowia* sp. (Y1D strain isolated from jellyfish) was considered because of its ability to directly impact the sensorial and technological properties of wines, since many strains are able to produce hydrolytic enzymes (glycosidases, proteases and pectinases) [75]. The selected strain of *Saccharomyces cerevisiae* LI 180-7 for this study had already demonstrated its ability for effective fermentation in difficult conditions, such as in the presence of high salt and phenol concentrations [35,62].

During fermentations controlled by bacterial and yeast strains, reducing sugars decreased, with the exception of samples inoculated with the bacterial *B. amyloliquefaciens* MS3 strain and the yeast *D. hansenii* BC T3-23 strain. These effects can be seen when, during fermentation, reducing sugars are produced by the decomposition of the starch material, and the content can increase if the production speed is faster than that used for microbial growth [76]. At the same time, the protein content reduced in all fermented Jellyfish paste samples throughout fermentation due to the decomposition of initial value [77]. Instead, the rise in the moisture content in the same samples can be interpreted as an increase in the amount of free water due to the polymer substances’ decomposition through the enzymes secreted by the microorganisms and the production of water through microbial metabolism [78]. Although a decrease in the initial total lipid content was detected in all the samples, fermented Jellyfish pastes can be considered a good source of lipids [79].

The use of the bacterial *B. amyloliquefaciens* MS3 strain and the yeast *D. hansenii* BC T3-23 strain maintained a high level of phenolic compounds and antioxidant activity, these results confirming that fermentation can help in maintaining the presence of bioactive phenolic compounds [80,81].

The acceptable performances demonstrated by the *B. amyloliquefaciens* MS3 revealed an interesting potential for this strain to be used as a starter for this sequential solid-state and submerged liquid fermentation. These data are in accordance with recent papers, where *B. subtilis* and *B. licheniformis* are considered as key contributors of soy sauce-like and soybean flavor [82], and in Korean traditional soy sauce, they revealed probiotic potential for humans and animals [83].

Regarding the nutritional traits, fermented Jellyfish pastes revealed interesting potential in terms of peptides, derived from the hydrolyzed proteins [84], and fatty acid profiles produced by the specific metabolic activities of the microbial strains that can be further explored. Furthermore, the phenolic content can be partially correlated to the detected antioxidant activity in the jellyfish-based fermented products.

Indeed, the total antioxidant activities (AA) were similar to the antioxidant content of foods of plant origins, known to contain high amounts of antioxidant compounds, when AA data were converted from Trolox equivalents to absolute values in mmol of electrons/hydrogen atoms donated in the redox reaction per 100 g of sample [85]. In fact, the highest AA value registered in the sample BC T3-23 (1.02 ± 0.02 mmol TE/100g FW, corresponding to 2.12 ± 0.03 mmol/100 g FW by the FRAP assay) was comparable to the antioxidant content found in fruits such as prunes and strawberries and in blueberry and cherry juice [85,86]. Moreover, the AA recorded in the samples MS3, KT 5-1 and SF (ranging from 0.53 to 0.64 mmol TE/100 g FW and corresponding to 1.04–1.25 mmol/100 g FW by FRAP assay) were comparable to the antioxidant content of cranberry juice, raspberry jam, grapefruit juice and pomegranate juice and fruits such as green apples, dried frigs, guava, green olives, oranges, pineapples and plums [85,86].

In the absence of established standards for these products, a preliminary test to recognize odor attributes of fermented Jellyfish pastes was carried out to identify the most promising samples characterized by acceptable flavor aspects. Voluntary panelists expressed their opinions on the fermented samples regarding the perceived odor impressions in order to identify the most promising treatment strategies for future analyses.

Recent studies reported sensory analyses of jellyfish produced by the traditional Asian method, treated in sodium chloride and alum and marinated using various seasonings [87,88]. Raposo et al. [13] for the first time described a procedure for cooking *Catostylus tagi* samples in a microwave oven and the sensory acceptance of jellyfish-based patès for preparing snacks.

Moreover, in this case, the presence of the bacterial *B. amyloliquefaciens* MS3 and the yeast *D. hansenii* BC T3-23 strains produced the highest scores of overall odor intensity and desirable odor classes, as umami, dried and red fruits and spices. As already observed by Jiang et al. [68], aroma-producing yeasts can be useful to generate characteristic flavor compounds in soy sauce, especially due to their ester biosynthesis pathways. The highest lipase activity of *D. hansenii* BC T3-23, which catalyzes the hydrolysis of acyl glycerols to fatty acids, di-acyl glycerols, mono-acyl glycerols and glycerol, can also be correlated to the esterase activity. Finally, the promising use of *B. amyloliquefaciens* as a starter is also proven by the ability to produce a more complex volatile compound profile in doenjang samples by strains selected for their multiple functional activities [89].

## 5. Conclusions

In this work, a completely new process for producing fermented jellyfish-based products (fermented Jellyfish pastes) was set up and optimized, starting from fresh samples of the jellyfish *Rhizostoma pulmo*, an abundant species in the Mediterranean Sea. All the data obtained in this study suggest that the bacterial *B. amyloliquefaciens* MS3 strain and the yeast *D. hansenii* BC T3-23 strain used as microbial starters can play important roles in fermented Jellyfish paste production. They can both drive and control the fermentation process in a manner similar to the traditional manufacturing method used for doenjang production [90] by controlling the pH, also favoring the production of LAB counts and desirable enzyme activities (amylases, lipases and proteases) strongly correlated to the development of preferred odor attributes (umami, dried and red fruits, spices), and maintaining adequate levels of nutritional traits such as total phenolic content and antioxidant activity comparable to several plant and fruit products known for their antioxidant content. Additional investigations of peptides and fatty acid profiles and digestibility assays are needed to attest the final product quality and the effects on human health of the fermented jellyfish paste-based formulation.

Although the data here reported are preliminary, the newly formulated jellyfish-based food fermented products can represent a new opportunity to stabilize a quickly perishable and potentially increasing marine resource, introduce edible jellyfish in Western countries and encourage the sustainable consumption of this new food source.

## Figures and Tables

**Figure 1 foods-11-03974-f001:**
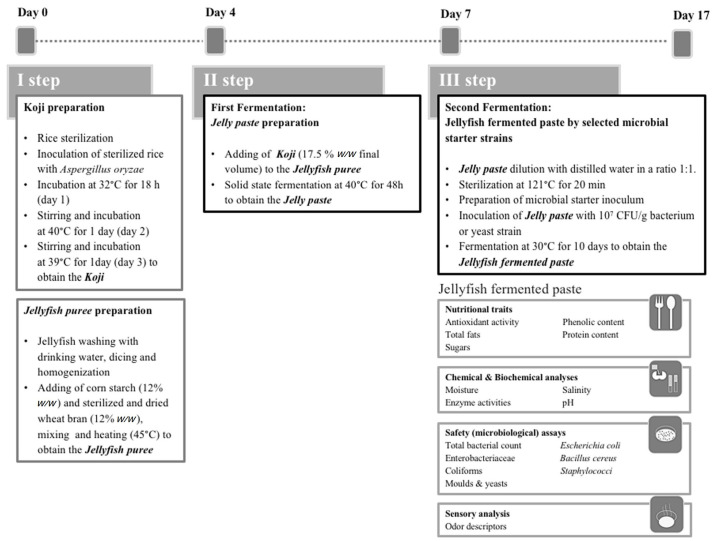
Diagram illustrating the procedure of the new proposed fermentation method for jellyfish-based food production. See also Appendix A.

**Figure 2 foods-11-03974-f002:**
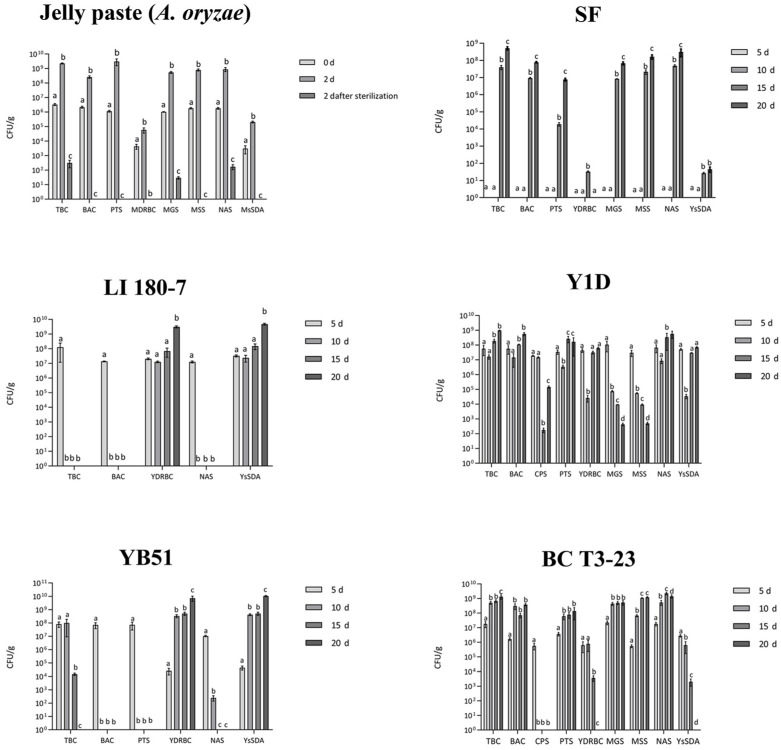
Evolution of microbial counts over 20 days of fermentation of Jellyfish-based food products inoculated with selected yeast strains. Jelly paste (*Aspergillus oryzae*): product obtained by the first fermentation step (solid-state fermentation) performed by inoculating the Koji starter (*A. oryzae*) into Jellyfish puree (consisting of jellyfish, starch and wheat bran). SF: spontaneous fermentation of uninoculated Jelly paste, as control. Y1D: *Metschnikowia* sp., YB51: *Candida parapsilosis*; BC T3-23: *Debaryomyces hansenii*; LI 180-7: *Saccharomyces cerevisiae*. The initial inoculum of the four yeast starter strains (LI 180-7, YB51, BC T3-23, Y1D) was about 10^7^ CFU/g. Microbial parameters: TBC (total bacterial count, accounting for aerobic colony count), BAC (*Bacillus* spp.), CPS (presumptive coagulase-positive staphylococci), PTS (presumptive total staphylococci), YDRBC (total yeast count), MDRBC (total mold count), MGS (MRS glucose seawater salts), MSS (MRS sucrose seawater salts), NAS (nutrient agar seawater salts). Media where microbial counts were equal to zero were not reported. For each fermented sample, microbial parameters were individually submitted to one-way analysis of variance (ANOVA), and Tukey’s post hoc method was applied to determine significant differences (*p* < 0.05) among the microbial counts at different time points (as showed in the graph legends).

**Figure 3 foods-11-03974-f003:**
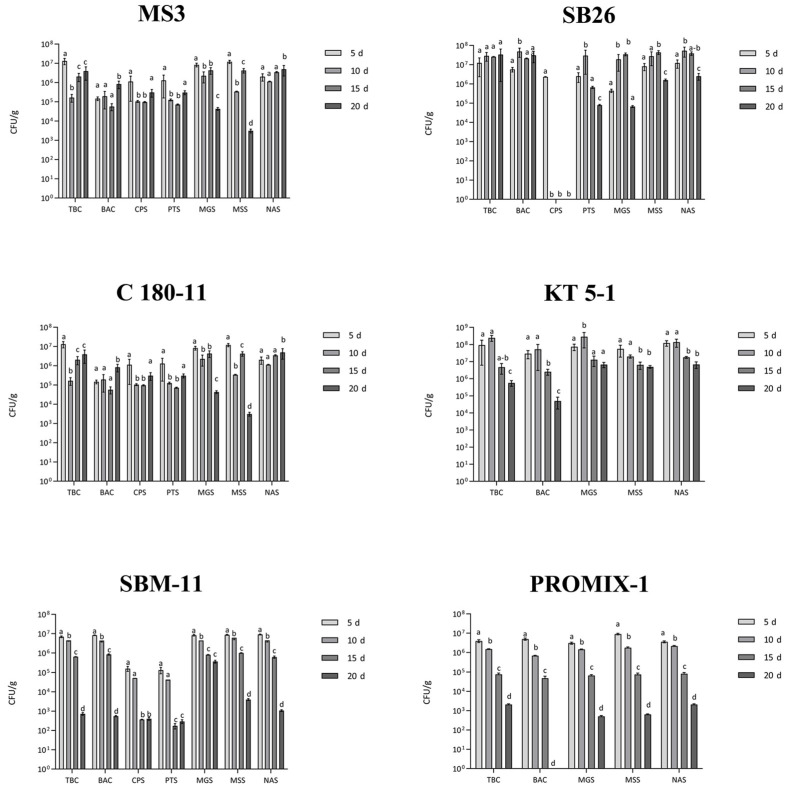
Evolution of microbial counts over 20 days of fermentation of Jellyfish-based food products (Jelly paste) inoculated with selected bacterial strains. SB26: *Staphylococcus pasteuri*; MS3: *Bacillus amyloliquefaciens*; C 180-11: *Lactiplantibacillus plantarum*; KT 5-1: *Leuconostoc mesenteroides*. Commercial microbial starter preparations provided by Sacco srl, SBM-11: *Lactobacillus sakei*, *Staphylococcus carnosus* and *S. xylosus*; PROMIX-1: *S. xylosus*. The initial inoculum of the four bacterial starter strains (C 180-11, KT-5-1, MS3, SB26) and of two commercial bacterial starter preparations (SBM-11, PROMIX-1, Sacco Srl, Cadorago, Italy) was about 10^8^ CFU/g. Microbial parameters: TBC (total bacterial count, accounting for aerobic colony count), BAC (*Bacillus* spp.), CPS (presumptive coagulase-positive staphylococci), PTS (presumptive total staphylococci), YDRBC (total yeast count), MDRBC (total mold count), MGS (MRS glucose seawater salts), MSS (MRS sucrose seawater salts), NAS (nutrient agar seawater salts). Media where microbial counts were equal to zero were not reported. For each fermented sample, microbial parameters were individually submitted to one-way analysis of variance (ANOVA), and Tukey’s post hoc method was applied to determine significant differences (*p* < 0.05) among the microbial counts at different time points (as showed in the graph legends).

**Figure 4 foods-11-03974-f004:**
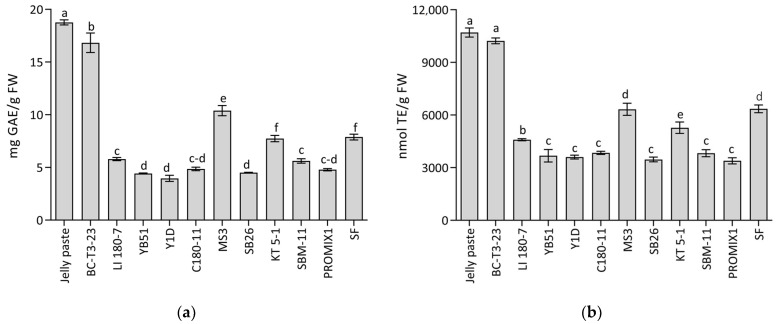
(**a**) Total phenolic contents and (**b**) antioxidant activity in fermented Jellyfish paste samples. Total phenolic content is expressed as mg gallic acid equivalents per g of fresh weight (GAE/ g FW); antioxidant activity is expressed as nmol of TE per g fresh weight (nmol TE/g FW). SF: spontaneous fermentation of uninoculated Jelly paste, as control. Jelly paste inoculated with yeast strains: Y1D, *Metschnikowia* sp.; YB51, *Candida parapsilosis*; BC T3-23, *Debaryomyces hansenii*; LI 180-7, *Saccharomyces cerevisiae*. Jelly paste inoculated with bacterial strains: SB26, *Staphylococcus pasteuri*; MS3, *Bacillus amyloliquefaciens*; C 180-11, *Lactiplantibacillus plantarum*, KT 5-1, *Leuconostoc mesenteroides*; SBM-11, *Lactobacillus sakei*, *Staphylococcus carnosus* and *Staphylococcus xylosus*; PROMIX-1, *Staphylococcus xylosus*. Data were submitted to one-way analysis of variance (ANOVA), Tukey’s post hoc method was applied to determine significant differences among samples (*p* < 0.05).

**Figure 5 foods-11-03974-f005:**
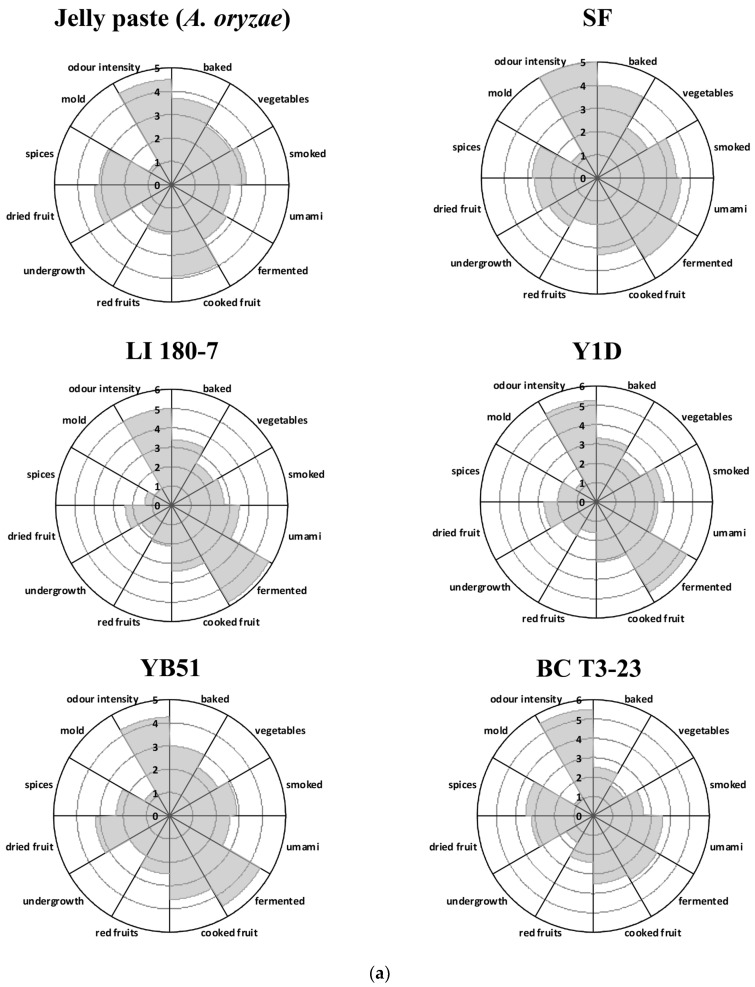
(**a**) Effects of fermentation by yeast starter cultures on odor characteristics of jellyfish-based food products. Jelly paste (*A. oryzae*): product obtained by the first fermentation step (solid-state fermentation) performed by inoculating the Koji starter (*A. oryzae*) into Jellyfish puree (consisting of Jellyfish, starch, and wheat bran). SF: spontaneous fermentation of uninoculated Jelly paste, as control; LI 180-7: *Saccharomyces cerevisiae*; Y1D: *Metschnikowia* sp.; YB51: *Candida parapsilosis*; BC T3-23: *Debaryomyces hansenii*. (**b**) Effects of fermentation by bacterial starter cultures on odor characteristics of jellyfish-based food products. MS3: *Bacillus amyloliquefaciens*; SB26: *Staphylococcus pasteuri*; C 180-11: *Lactiplantibacillus plantarum*, KT 5-1: *Leuconostoc mesenteroides*. SBM-11: *Lactobacillus sakei*, *Staphylococcus carnosus* and *S. xylosus*; PROMIX-1: *S. xylosus*.

**Figure 6 foods-11-03974-f006:**
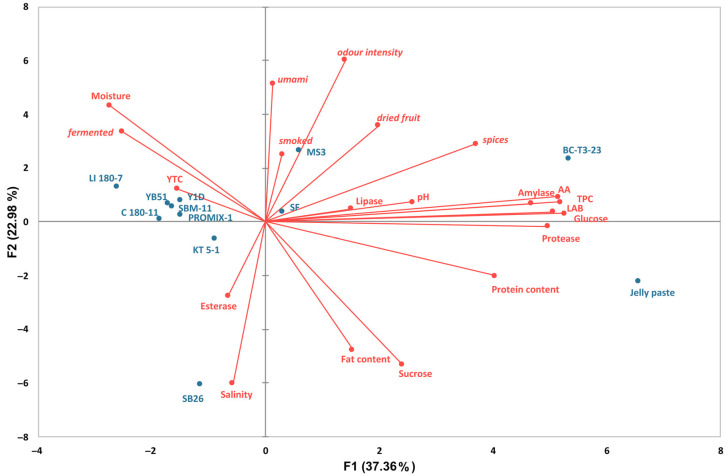
Score scatter plot of PCA model performed on parameters associated with all jellyfish-based fermented samples. PCA variables were the data obtained from the analysis of values of microbiological data, chemical composition, nutritional traits, enzyme-associated activities and odor descriptors at the end of the process.

**Table 1 foods-11-03974-t001:** Bacterial isolates identified from *Rhizostoma pulmo* jellyfish samples. Bacterial strains were identified according to their similarity to 16S rDNA region sequences in the GenBank nucleotide sequence database. The strains and the nucleotide sequence of their rDNA regions were deposited in the NCBI database; the corresponding accession numbers are given.

Strain Code (GenBank Accession Number)	Identification *	Identity	Closest Match Accession Number
FB6 (OP555308) FB11 (OP555310) LB13 (OP555314) LB17 (OP555313) LB18 (OP555312) LB24 (OP555311)	*Staphylococcus pasteuri*	100.00%	MG757632.1
LB11 (OP555309) LA52 (OP555315)	*Bacillus simplex*	100.00%, 99.82%	MT310831.1, LT547805.1
LA30 (OP555316)	*Bacillus amyloliquefaciens*	100.00%	CP054479.1
LA2 (OP555320) LA35 (OP555317)	*Enterobacter cloacae*	99.78%, 99.65%	MT367828.1, KP226569.1
LA6 (OP555319) LA27 (OP555318)	*Leclercia adecarboxylata*	99.79%	MT180610.1
LA7 (OP555292)	*Enterobacter hormaechei* subsp. *xiangfangensis*	99.66%	CP030007.1
FB1 (OP555302) FB4 (OP555295) FB5 (OP555294) FB14 (OP555293) GB2 (OP555300) GB4 (OP555299) GB7 (OP555298) GB9 (OP555297) GB10 (OP555296)	*Staphylococcus pasteuri*	100.00%	CP031280.1
GB1 (OP555301)	*Staphylococcus epidermidis*	100.00%	CP040883.1
SB18 (OP555307) SB19 (OP555306) SB20 (OP555305) SB24 (OP555304) SB26 (OP555303)	*Staphylococcus pasteuri*	100.00%	MT072161.1

* When the identity percentage of the obtained sequences was <99%, the isolates were identified only at genus level.

**Table 2 foods-11-03974-t002:** Yeast isolates identified from *Rhizostoma pulmo* jellyfish samples. Yeast strains were identified according to their similarity to ITS1-5.8S-ITS2 region sequences in the GenBank nucleotide sequence database. The strains and the nucleotide sequence of the ITS1-5.8S-ITS2 region were deposited in the NCBI database; the corresponding accession numbers are given.

Strain Code (GenBank Accession Number)	Identification *	Identity	Closest Match Accession Number
YD19 (OP554736)	*Metschnikowia* sp.	98.25%	NR_166218.1
Y1D (OP554737)	*Metschnikowia* sp.	96.20%	MK394155.1
YC18 (OP554738)	*Metschnikowia* sp.	95.99%	KY495746.1
YB51 (OP554739)	*Candida parapsilosis*	99%	MH545914.1
YLq1 (OP554740)	*Aureobasidium pullulans*	100.00%	MT035961.1
YB1 (OP554741)	*Rhodotorula diobovata*	100.00%	KY104778.1

* When the identity percentage of the obtained sequences was <99%, the isolates were identified only at genus level.

**Table 3 foods-11-03974-t003:** Main parameters chosen for safety analysis of fermented Jellyfish paste samples.

Assays	Limits	Analytical Reference Method	Reference
Aerobic colony count (total bacterial count, TBC)	<10^4^ CFU/g (for salami, cured meats, ham, etc.) 10^5^ < X < 10^7^ CFU/g (for fish-based products)	UNI EN ISO 4833-1:2013 [47,48]	[49,50]
β-glucuronidase-positive *Escherichia coli*	<10 CFU/g	ISO 16649-2:2001	[47,48]
Enterobacteriaceae	<10^2^ CFU/g		[51]
Presumptive *Bacillus cereus*	<10^2^ CFU/g		[51]
Coliforms	<10 CFU/g <=70MPN/100 mL	ISO 4382:2006	[50]
Coagulase-positive staphylococci	<10^2^ CFU/g, 10^2^ < X < 10^3^ CFU/g	UNI EN ISO 6888-2:1999	[47,48]
Molds and yeasts	<10^2^ CFU/g (marinated octopus, seafood cocktail)	ISO 21527-1:2008	[49]

**Table 4 foods-11-03974-t004:** Effect of selected microbial starter cultures on fermented Jellyfish paste samples.

Sample	Moisture mg/g	Fat mg/g FW	Glucose mg/g FW	Sucrose mg/g FW	Protein mg/g FW
Jelly paste	713.74 ± 31.08 (a)	128.53 ± 10.33 (a)	3.20 ± 0.02 (a)	0.22 ± 0.02 (a)	10.58 ± 0.33 (a)
Jelly fermented paste with yeast starter					
BC T3-23	877.57 ± 44.48 (b)	85.22 ± 5.96 (b)	2.93 ± 0.013 (b)	ND	1.42 ± 0.02 (b)
LI 180-7	923.44 ± 6.81 (b)	40.97 ± 3.06 (c)	0.57 ± 0.002 (c)	0.012 ± 0.001 (b)	0.75 ± 0.09 (c)
YB51	928.80 ± 54.42 (b)	79.05 ± 5.05 (b)	0.52 ± 0.01 (d)	ND	0.51 ± 0.06 (c)
Y1D	858.58 ± 1.47 (b)	112.13 ± 9.57 (a)	0.53 ± 0.004 (d)	0.006 ± 0.001(b)	0.55 ± 0.02 (c)
Jelly fermented paste with bacterial starter					
C180-11	825.66 ± 12.33 (b)	75.98 ± 6.05 (b)	0.51 ± 0.01 (d)	0.01 ± 0.002 (b)	0.91 ± 0.18 (b, c)
MS3	934.43 ± 46.89 (b)	19.89 ± 2.10 (d, c)	1.54 ± 0.002 (e)	ND	0.72 ± 0.06 (b)
SB26	803.45 ± 21.49 (b)	169.48 ± 9.88 (e)	0.75 ± 0.004 (f)	0.19 ± 0.002 (c)	0.74 ± 0.11 (c)
KT 5-1	834.27 ± 33.67 (b)	62.57 ± 4.44 (f, b)	0.59 ± 0.004 (c)	0.01 ± 0.002 (b)	1.42 ± 0.07 (c)
Jelly fermented paste with bacterial commercial starter					
SMB-11	953.51 ± 41.93 (b)	37.92 ± 1.38 (c)	0.50 ± 0.004 (d)	0.03 ± 0.005 (d)	0.63 ± 0.09 (c)
PROMIX-1	863.45 ± 22.85 (b)	42.55 ± 4.54 (c)	0.51 ± 0.001 (d)	0.01 ± 0.004 (b)	0.26 ± 0.04 (c)
Jelly fermented paste without starter					
SF	832.59 ± 8.97 (b)	142.06 ± 2.03 (a)	0.94 ± 0.02 (g)	0.11 ± 0.004 (e)	1.52 ± 0.04 (b)

ND: not detected. The different letters in a line indicate significant differences between samples (*p* < 0.05).

**Table 5 foods-11-03974-t005:** Evolution of chemical–physical parameters during the fermentation of Jellyfish paste samples.

Sample	pH				Temperature				Salinity (%)			
	Time (Days)				Time (Days)				Time (Days)			
	5	10	15	20	5	10	15	20	5	10	15	20
BC T3-23	5.26	5.4	5.19	5.39	28	24.2	20	23.5	15	16	16	16
LI 180-7	4.85	4.905	4.68	4.9	26.1	24.1	22.7	22.7	15	16	16.6	16
YB51	5.05	5.195	5.01	5.06	27.6	24.4	23.1	23.2	14	15	15	14
Y1D	5.06	4.645	4.61	4.77	28.1	23.6	22.3	22.5	14	15	15	14
C 180-11	3.4	3.505	3.18	3.23	27.7	23.7	22.1	23	17	18	19	18
MS3	5.3	5.1	5.88	5.17	28.3	23.6	23.3	23.3	14	15	14.4	14
SB26	5.12	4.985	4.71	4.69	27.6	24.1	22	22.3	16	16	17	17
KT 5-1	4.22	4.37	4.1	4.15	26	23.1	22.8	22.3	14	18	18	19
SBM-11	3.5	3.46	3.47	3.38	23.5	24	21.9	23.2	16	16	16	17
PROMIX-1	3.42	3.2	3.66	3.37	23.1	24.2	21.7	27	16	17	17	17
SF	5.23	5.195	4.77	5.08	29.2	23.9	22.1	22.04	13	18	17.5	15

**Table 6 foods-11-03974-t006:** Enzyme activities associated with fermented Jellyfish paste samples.

	Lipase U/g	Esterase mU/g	Amylase U/g	Protease U/g
Jelly paste *	152.16 ± 3.50 (a)	20.54 ± 0.62 (a)	53.03 ± 0.68 (a)	482.23 ± 4.11 (a)
Jelly fermented paste with yeast starter				
BC T3-23	418.31 ± 4.94 (b)	7.63 ± 0.47 (b)	73.94 ± 0.93 (b)	481.02 ± 5.68 (a)
LI 180-7	220.32 ± 5.07 (c)	20.83 ± 0.46 (a)	5.88 ± 0.74 (c)	165.06 ± 5.60 (b)
YB51	85.57 ± 4.64 (d)	12.39 ± 0.48 (c)	18.42 ± 0.69 (d)	114.90 ± 2.42 (c)
Y1D	207.91 ± 5.29 (e)	19.47 ± 0.38 (d)	25.34 ± 0.20 (d)	162.09 ± 6.43 (b)
Jelly fermented paste with bacterial starter				
C 180-11	172.14 ± 2.19 (f)	12.36 ± 0.52 (e)	5.97 ± 0.70 (c)	103.75 ± 1.72 (c)
MS3	185.06 ± 5.73 (g)	10.40 ± 0.36 (f)	17.14 ± 0.61 (d)	148.49 ± 0.38 (b)
SB26	249.09 ± 4.38 (f)	17.34 ± 0.16 (g)	17.68 ± 0.43 (d)	166.81 ± 6.26 (b)
KT 5-1	220.91 ± 4.61 (c)	18.16 ± 0.38 (h)	16.65 ± 0.60 (d)	193.70 ± 6.21 (d)
Jelly fermented paste with bacterial commercial starter				
SBM-11	165.60 ± 5.75 (h)	16.59 ± 0.03 (g)	23.88 ± 0.74 (d)	198.11 ± 4.66 (d)
PROMIX-1	273.88 ± 4.91 (i)	9.44 ± 0.06 (f)	24.29 ± 0.19 (d)	107.27 ± 6.28 (c)
Jelly fermented paste without starter				
SF	51.00 ± 0.29 (l)	6.27 ± 0.54 (i)	17.26 ± 0.35 (d)	152.35 ± 3.98 (b)

* Jelly paste: sample obtained after *Aspergillus oryzae* treatment and before sterilization for the subsequent submerged fermentation step. The different letters in line indicate significant differences between samples (*p* < 0.05).

**Table 7 foods-11-03974-t007:** Odor descriptors selected for fermented jellyfish paste samples.

Olfactory	Reference Odor
Desired perceptions	
Umami	miso, soy sauce, barbecue sauce, stock cube
Dried fruits	almond, dried fig, dry plum
Red fruits	blue raspberry, soft fruits, cherry
Spices	vanilla, liquorice, laurel, rosemary
Neutral perceptions	
Vegetables	pepper, eggplant, artichoke, tomato
Cooked fruit	baked apple, baked fruit, ripe fruit, quince jam
Undesired perceptions	
Baked	bread, bread crust, bran, legumes
Smoked	roast meat, speck, frankfurter
Fermented	beer, yeast, sourdough, must, wine, fermented fruit
Undergrowth	mushroom, herbal, pungent
Mold	moldy
Overall	
Odor intensity	global intensity of the odor

## Data Availability

Relevant data is contained within the article or Appendix A. Additional data are avaliable from the corresponding author.

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
