# Peer review of "Combination of Solid State and Submerged Fermentation Strategies to Produce a New Jellyfish-Based Food"

_foods, 2022, doi:10.3390/foods11243974_

Round 1
Reviewer 1 Report
This manuscript design a new process for producing fermented jellyfish-based products (Jellyfish fermented pastes) , starting from fresh samples of the jellyfish Rhizostoma pulmo with corn starch 12% (w/w) and dried wheat bran 12% (w/w), first step is a solid-state fermentation by inoculated the rice koji containing the starter A. oryzae , and the second one is a submerged fermentation conducted by selected bacterial and yeasts strains deriving from different food sources, Nutritional analyses was carried out by analyzing reducing sugars, total lipids, proteins and total phenols content, and antioxidant activity in all the samples,The new fermented products obtained by the use of Debaryomyces hansenii BC T3-23 yeast strain and the Bacillus amyloliquefaciens MS3 bacterial strain revealed good nutritional traits in terms of protein, lipids and total phenolic content, as well as a valuable total antioxidant activity. This work have some creative, but some selected quality indicators needs to be questioned. My revision opinions are as follows:
1. The manuscript should give us the chemical composition of selected jellyfish, such as Individual content of protein, lipid and carbohydrate,
2. Jellyfish fermented pastes should have a important indicator of amino nitrogen content
3. total phenols come from which material?
4. If Jellyfish fermented pastes was used as a commercial food, it must be sterilized, the amylase, protease and lipase enzyme will be deactivation, complex enzyme will be meaningless. Odorous descriptors will also change.
Author Response
1. The manuscript should give us the chemical composition of selected jellyfish, such as Individual content of protein, lipid and carbohydrate,
We agree with the Reviewer, thanks for the suggestion. In order to give a more complete view of the data, the following sentence was added to line …493-497, pag. 15.
“The characterization of fresh jellyfish as raw material was performed and the results were in the range of values already found by previous works [9, 16, 19]. In particular, the total proteins were in the range from 2.4 to 4 mg/g of fresh weight, lipids were in the range of 0.2-0.3 mg/g of fresh weight, while carbohydrates represented minor and often negligible components of the jellyfish tissues.”
2. Jellyfish fermented pastes should have a important indicator of amino nitrogen content
The amino nitrogen content was not evaluated in this work. In our experience the protein content is often overestimated when calculated as “crude protein” = 6.25 × nitrogen content (N) mainly in fresh material. Then, it is generally preferable to carry out a dosage of proteins and/or amino acids which are a more reliable indicator of the total protein content. Anyway, if the reviewer suggests to consider this parameter in order to follow the enrichment in proteins of the fermented product, or protein content modification, the reviewer is right and we can consider this point in the following work. We hope the data presented here can be enough for this very preliminary work on the fermented jellyfish
3. total phenols come from which material?
The values of total phenols in raw material consisting of the fresh jellyfish are indicated in the text (see new text at pag 17 lines 554-570), anyway we cannot consider jellyfish the only origin of phenol compounds as they were not measured in the others ingredients and they could also derive from fermentation processes.
Please see pag 17 lines 553-569
“The content of total phenols in fresh R. pulmojellyfish was in the range of 45 - 60 μg GAE/g of fresh weight in agreement with previous results [11]. A decrease of the phenol content in Jelly paste and in all the bacteria and yeast inoculated Jelly fermented paste samples was observed (Figure 4a) also in comparison with the Koji (A. oryzae) treated Jelly paste (Figure 4a). However, the samples inoculated with the bacterial strain MS3, and the yeast strain BC T3-23 maintained the highest levels of these compounds, followed by one treated with the bacterial strain KT 5-1. In this last case, the phenolics content was comparable with the SF sample.
A similar trend to the phenol content was observed in the antioxidant activity (AA), expressed as nmol of Trolox Equivalent (TE) per gram of FW. Considering that the antioxidant activity in raw untreated material of R. pulmowas about 800 ± 164 nmol TE/g FW in agreement with Leone et al. [11], an increase of AA was detected in Jelly fermented paste, probably due to the fermentation metabolism with the formation of antioxidant metabolites. This is also in agreement with De Domenico et al. [15], where the digestion of proteins produced small peptides with higher antioxidant activity than undigested proteins. The antioxidant activity can be related to various chemical species that are formed during the fermentation processes”.
- If Jellyfish fermented pastes was used as a commercial food, it must be sterilized, the amylase, protease and lipase enzyme will be deactivation, complex enzyme will be meaningless. Odorous descriptors will also change.
We understand the reviewer’s criticism, however, this study is the first one conducted by using the raw JF as ingredient for food fermentation and it is an explorative work that can offer new inputs for future applications. The jelly paste fermented products are not commercially available and they are a novel food. Then, the manner in future they can be used as a food for humans have still to be described. At the present time, they can be considered not as a food by themselves but also as an ingredient for other food applications since they can offer a source of enzymes.
We agree also that odour descriptor can be modified by a possible heat treatment, but our scope in this preliminary study was to firstly evaluate the possible contribution of various microbial starters to the odour notes and to evaluate a general acceptability of the product, by excluding at this stage off-flavours and smells.
Reviewer 2 Report
Dear Authors, please find here my comments.
P1L34 (Page1 Line 34). "...has not already applied to..." perhaps is better to write "has not yet been applied to..."
P1L41-42. Please write this instead: "Indeed, jellyfish is regarded as a high-quality diet marine product."
P2L49. Please revise the sentence: "Many studies have concerned the instant jellyfish in China".
P4L183. "... to ensure the thermal exchanges." what thermal exchanges? isn't for gas transfer/aeration?
P5L197: "... strains isolated and identified by R. pulmo..." please write instead "... strains isolated and identified from R. pulmo..."
Please, it is not clear is you have isolated those bacteria and yeast in a previous study, or if that isolation was part of your present study? If it was part of a previous study, please give the refences. If it was part of your present study, please add the isolation methods in the Materials and Methods section, and add the isolation results in the Results section.
P6L218. How did you measured pH?
P8L340: "In this study the main goal was the set up and optimization of a fermentation strategy applied to a composite raw material containing as principal ingredient jellyfish biomass" This goal is not directly expressed in your Abstract, nor Introduction. Please make sure to state clearly there, abstract and introduction, what your main goal/objective was.
P11,12, Figure 2 and 3. Dear authors, the font on the figure is too small. Please, if possible reorganize the plots having 2 columns by 3 rows, instead of the 3 columns by 2 rows.
P17, Figure 17. Similar situation as in Figure 3. Please reorganize both subplot sections a and b. Subplot section a. 2 columns by 3 rows. Subplot section b. 2 columns by 3 rows. I hope that will make your plot figures bigger hence the font will be easy to read.
P15, Figure 4. Total phenolic and antioxidant activity in Jellyfish. Do you have the results for the unfermented jellyfish? if yes please add those results here. If not, please write that you have not perform those tests.
P21L770-776. Please in the conclusions only write conclusions, not descriptions of your study.
End of review.
Author Response
Dear Authors, please find here my comments.
P1L34 (Page1 Line 34). "...has not already applied to..." perhaps is better to write "has not yet been applied to..."
Thanks to the reviewer. The sentence was improved (please see Pag 1 Line 37)
P1L41-42. Please write this instead: "Indeed, jellyfish is regarded as a high-quality diet marine product."
Thanks to the reviewer. The sentence was improved (please see Pag 1 Line 44)
P2L49. Please revise the sentence: "Many studies have concerned the instant jellyfish in China".
Thanks to the reviewer. The sentence was changed ((please see Pag 2 Line 52)
P4L183. "... to ensure the thermal exchanges." what thermal exchanges? isn't for gas transfer/aeration?
Yes, this is the correct interpretation, thank you. The sentence was modified (please see Pag 5 Line 197)
P5L197: "... strains isolated and identified by R. pulmo..." please write instead "... strains isolated and identified from R. pulmo..."
Thanks to the reviewer. The sentence was changed (please see Pag 5 Line 210-211)
Please, it is not clear is you have isolated those bacteria and yeast in a previous study, or if that isolation was part of your present study? If it was part of a previous study, please give the refences. If it was part of your present study, please add the isolation methods in the Materials and Methods section, and add the isolation results in the Results section.
The text was improved by adding the isolation methods and the results in the related sections.
See Lines …144-152, pages 3-4
“The isolation of yeast and bacteria from jellyfish tissue were performed following the procedure already described by Bleve et al. [9, 10]. Based on morphological traits, representative isolated bacteria and yeast were selected. The isolated bacteria were maintained on saline MRS Agar or saline PCA added with 0.05 g/L nystatin Sigma-Aldrich, Darmstadt, Germany). The yeast isolated were maintained on saline Sabouraud Dextrose Agar (Sigma-Aldrich, Darmstadt, Germany) supplemented with 0.05 g/L ampicillin (Sigma-Aldrich, Darmstadt, Germany) at 28 °C for 2-3 days. Bacteria maintained on saline MRS agar were grown anaerobically at 28°C for 3-4 days. Bacteria maintained on saline PCA were grown aerobically at 28 °C for 2–3 days.”
and Lines 375-381 pages 9-10 .
“…on saline media.
A total number of 180 bacterial colonies were randomly selected, according to their different morphological aspects, from saline PCA and a number of 52 colonies from saline MRS. Concerning yeasts, a total of 24 isolates were selected onto saline Sabouraud Dextrose Agar medium. According to their different morphological aspects, 29 bacterial colonies from saline PCA and saline MRS agar medium and 6 yeast colonies were randomly selected”.
P6L218. How did you measured pH?
The following sentence was introduced. Lines…237-239, page 6..
pH was measured using a pH-meter (HANNA INSTRUMENTS Italia srl - Ronchi di Villafranca Padovana, Padova, Italy).
P8L340: "In this study the main goal was the set up and optimization of a fermentation strategy applied to a composite raw material containing as principal ingredient jellyfish biomass" This goal is not directly expressed in your Abstract, nor Introduction. Please make sure to state clearly there, abstract and introduction, what your main goal/objective was.
We thank the reviewer. Both the abstract (Line 13-14) and introduction (lines 98-100) were improved, as suggested.
P11,12, Figure 2 and 3. Dear authors, the font on the figure is too small. Please, if possible reorganize the plots having 2 columns by 3 rows, instead of the 3 columns by 2 rows.
Thank you for the suggestion. The Figures 2 and 3 were reorganized
P17, Figure 17. Similar situation as in Figure 3. Please reorganize both subplot sections a and b. Subplot section a. 2 columns by 3 rows. Subplot section b. 2 columns by 3 rows. I hope that will make your plot figures bigger hence the font will be easy to read.
Thank you for the suggestion. Figure 5 was modified. We hope now it is easier to read.
P15, Figure 4. Total phenolic and antioxidant activity in Jellyfish. Do you have the results for the unfermented jellyfish? if yes please add those results here. If not, please write that you have not perform those tests.
Thank you to the Reviewer for pointing out this important issue. Analyses of the total phenols and antioxidant activity were performed in the present and previous works by our group. The obtained values and related text were added at pag…17..lines…553-569.
“The content of total phenols in fresh R. pulmo jellyfish was in the range of 45 - 60 μg GAE/g of fresh weight in agreement with previous results [11]. A decrease of the phenol content in Jelly paste and in all the bacteria and yeast inoculated Jelly fermented paste samples was observed (Figure 4a) also in comparison with the Koji (A. oryzae) treated Jelly paste (Figure 4a). However, the samples inoculated with the bacterial strain MS3, and the yeast strain BC T3-23 maintained the highest levels of these compounds, followed by one treated with the bacterial strain KT 5-1. In this last case, the phenolics content was comparable with the SF sample.
A similar trend to the phenol content was observed in the antioxidant activity (AA), expressed as nmol of Trolox Equivalent (TE) per gram of FW. Considering that the an-tioxidant activity in raw untreated material of R. pulmo was about 800 ± 164 nmol TE/g FW in agreement with Leone et al. [11], an increase of AA was detected in Jelly fermented paste, probably due to the fermentation metabolism with the formation of antioxidant metabolites. This is also in agreement with De Domenico et al. [15], where the digestion of proteins produced small peptides with higher antioxidant activity than undigested proteins. The antioxidant activity can be related to various chemical species that are formed during the fermentation processes.”
P21L770-776. Please in the conclusions only write conclusions, not descriptions of your study.
Thanks to the reviewer. The Conclusion section was amended.